# Nanomanufacturing of silicon surface with a single atomic layer precision via mechanochemical reactions

Lei Chen [1], Jialin Wen [2], Peng Zhang[1], Bingjun Yu[1], Cheng Chen[1], Tianbao Ma[2], Xinchun Lu[2], Seong H. Kim [1,3] & Linmao Qian [1]

Topographic nanomanufacturing with a depth precision down to atomic dimension is of importance for advancement of nanoelectronics with new functionalities. Here we demonstrate a mask-less and chemical-free nanolithography process for regio-specific removal of atomic layers on a single crystalline silicon surface via shear-induced mechanochemical reactions. Since chemical reactions involve only the topmost atomic layer exposed at the interface, the removal of a single atomic layer is possible and the crystalline lattice beneath the processed area remains intact without subsurface structural damages. Molecular dynamics simulations depict the atom-by-atom removal process, where the first atomic layer is removed preferentially through the formation and dissociation of interfacial bridge bonds. Based on the parametric thresholds needed for single atomic layer removal, the critical energy barrier for water-assisted mechanochemical dissociation of Si–Si bonds was determined. The mechanochemical nanolithography method demonstrated here could be extended to nanofabrication of other crystalline materials.

[1] Tribology Research Institute, State Key Laboratory of Traction Power, Southwest Jiaotong University, Chengdu 610031, China. [2] State Key Laboratory of Tribology, Tsinghua University, Beijing 100084, China. [3] Department of Chemical Engineering and Materials Research Institute, The Pennsylvania State University, University Park, Pennsylvania 16802, USA. Correspondence and requests for materials should be addressed to X.L. (email: xclu@tsinghua.edu.cn) or to S.H.K. (email: shkim@engr.psu.edu) or to L.Q. (email: linmao@swjtu.edu.cn)

Nanomanufacturing process with an ultra-high precision is of paramount importance for new development of nanoelectronics with unique functionalities[1–3]. The ultimate precision that can be achieved on a crystalline substrate could be defined as the topographic control down to a single atomic layer. In addition, the topographic patterning should be done at a specific location with an arbitrary shape without causing subsurface damages or disorders.

The most widely used nanomanufacturing method is etching-based lithography which uses wet chemicals or high-energy plasma[4–6]; however, it is often difficult to control reaction kinetics down to the single atomic level. Recently, atomic layer etching (ALE) was demonstrated[7]. Similar to the well-known atomic layer deposition (ALD) process, ALE relies on sequential self-limiting thermal reactions; the only difference is that the final result is removal of a single atomic layer, instead of deposition. All these methods require sacrificial masks for regio-selective patterning, which involve additional processing steps.

Various mask-free and chemical-free lithographic methods have been demonstrated in the literature. Conventional micromachining processes, such as diamond turning, are an early example, but the thickness control in material removal is only on the order of nanometer, much thicker than the dimension of atomic layers[8,9]. Nanoimprint and focused ion beam (FIB)-assisted nanolithography can realize much higher precision; however, the destructions of subsurface crystalline structures are often accompanied[10,11].

More recently, scanning probe lithography (SPL) has been employed as an alternative means for high-precision material removal or modification[12–23]. Examples include localized deposition of organics through capillary flow (such as known as dip-pen lithography)[16,17], local deposition of polymer melts with a heated probe (which forms glassy organic resist upon cooling)[18,19], and localized electrochemical oxidation forming $SiO_2$ masks[20,21] or reduction of graphene oxide at the nanoscale[22,23]

using an electrically-biased tip. In these approaches, the patterns produced with SPL can act as masks and the transfer of such patterns to the silicon substrate requires subsequent etching processes.

As a means of chemical-free process, abrasive wear can be employed and controlled to the atomic scale with SPL. It has been demonstrated for mica, graphite, and KBr[24–27]; but, this approach has not been demonstrated for silicon surfaces which play pivotal roles in semiconductor and optics industries[3,28]. Direct nano-patterning on a silicon surface was recently reported, which utilized scanning tunneling microscopy (STM) to induce the electron-stimulated desorption of hydrogen from the hydrogen-passivated silicon wafer; this allows selective reaction of functional groups at those sites[29]. But, this method cannot remove silicon atoms directly producing topographic features.

Here, we demonstrated SPL tip-assisted, mask-less, and chemical-free nanolithography to attain direct etching of a single crystalline silicon (Si) wafer with a depth control down to the ultra-precision level—single atomic layer. The process is based on shear-induced mechanochemical reactions (also called tribochemical reactions) carried out in an ambient condition with a controlled humidity. Different from mechanical wear involving abrasion, fracture, or plastic deformation, this method relies on the shear-induced hydrolysis reaction of silicon with adsorbed water molecules which are in equilibrium with water vapor in the gas phase. High resolution transmission electron microscopy (HRTEM) analysis provided the direct evidence of the outermost atomic layer removal on the Si(100) surface without subsurface structural damage or defect. The molecular details of mechanochemical wear via atom-by-atom removal processes were revealed with molecular dynamics (MD) simulations. This technique could be applied for topographic fabrication with an atomic depth resolution for other materials that can undergo shear-induced mechanochemical reactions.

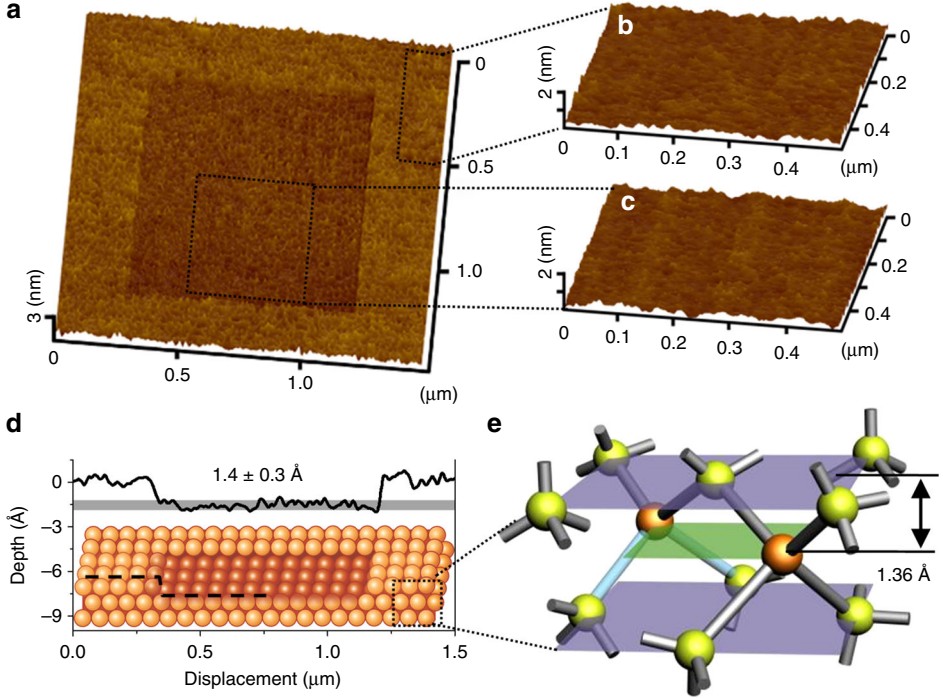

**Fig. 1** Single atomic layer removal of Si material. **a** SPM image (1.5 μm × 1.5 μm) of the manufactured area. Topographic images (0.5 μm × 0.5 μm) of **b** the original surface, and **c** the manufactured surface. **d** Cross-section profile of the manufactured area corresponding to the single atomic layer removal on Si (100). **e** Crystal structure of Si(100)

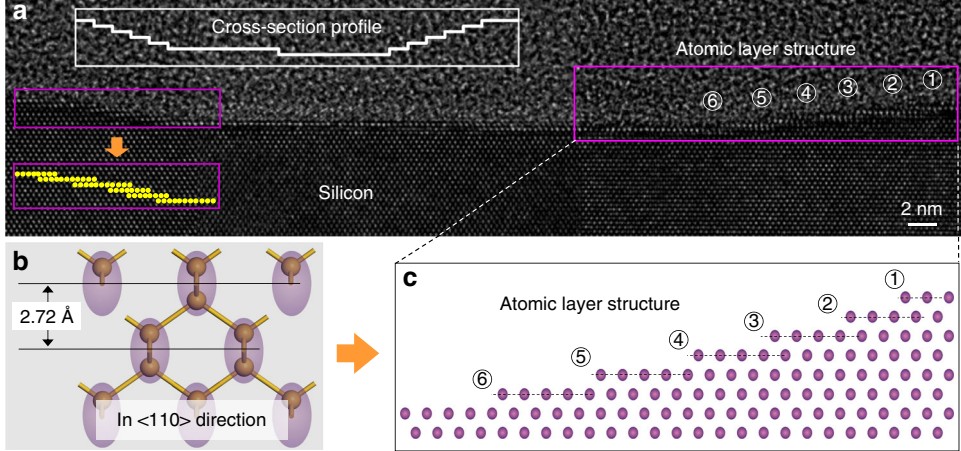

**Fig. 2** Evidence for no subsurface damage in the mechanochemically-etched region. **a** Cross-section TEM image of mechanochemical wear region. A visual aid for atomic steps in the left edge is added with yellow dots. **b** Atomic structure model of Si(100) in the <110> direction. **c** Visualization of the atomic step edges in the right edge region of the wear scar. Each circle in **c** corresponds to the dimer marked with the purple ellipse in **b**

## Results

**Single atomic layer removal of Si(100).** The mechanochemical nanofabrication process on a Si(100) surface was conducted with scanning probe microscopy (SPM) using a silica microsphere probe (Supplementary Fig. 1) in ambient air with a relative humidity (RH) of 75% ± 2%. The scanned area was imaged with a sharp $Si_3N_4$ tip in vacuum. The region scanned once with the silica sphere at an applied load $F_n$ of 300 nN (the contact stress calculated with the DMT contact mechanics model ($\sigma$) was 247 MPa) showed a 1.4 ± 0.3 Å deep depression in topographic imaging (Fig. 1d). On the Si(100) surface, the theoretical thickness of monolayer is 1.36 Å (Fig. 1e), which is very close to the observed depth within the accuracy of SPM (Supplementary Fig. 2). In humid air, the presence of adsorbed water enables the mechanical shear-induced hydrolysis reactions of the substrate atoms exposed at the surface[30–32]. The number of atomic layer removed on Si (100) strongly depends on the mechanical pressure applied by the silica sphere. No surface wear was observed at a contact stress below 247 MPa (Supplementary Fig. 3); thus, this value could be taken as an upper bound of the critical contact stress for mechanochemical removal of a single layer of silicon atoms under the given scanning speed and RH conditions.

Although the SPM imaging with the $Si_3N_4$ tip cannot visualize the Si surface with an atomic resolution due to the instability of the cantilever with a low spring constant (0.1 N m$^{-1}$), it is noted that the root-mean-square (RMS) roughness inside the 500 × 500 nm² scanned area (0.11 nm) (Fig. 1b) is similar to that of the original Si surface (0.10 nm) (Fig. 1c). However, the surface roughness alone cannot tell if the subsurface atomic order of the crystalline silicon structure is conserved or not. So, cross-sectional imaging with HRTEM was carried out to check if there is any subsurface plastic flow or fracture.

**Atomic layer removal characterized by transmission electron microscopy (TEM) observation.** To confirm the absence of subsurface structural damages after the topmost layer removal, a shallow line feature was produced and a thin cross-section of the line feature region was carved out using FIB milling. The wear scar with a depth of around 16 Å was formed by scanning the $SiO_2$ sphere at a contact stress $\sigma$ of 571 MPa ($F_n$ was 2.3 μN) in a line-scratch mode (Supplementary Fig. 4). The lattice-resolved TEM image shown in Fig. 2 manifests that the Si atoms beneath the line-scanned area keep the perfect crystallographic order even in the outermost exposed surface layer. Thus, mechanical wear,

plastic deformation via phase transformation, and lattice defect formation (dislocation and slipping) can be ruled out. The amorphization beneath the topographically-worn area did not occur. Furthermore, the atomic steps revealed at both edges of the sliding tract (Fig. 2a, c, and Supplementary Fig. 5c and d) are consistent with the atom-by-atom wear model[33–35]. When the contact stress is reduced to 247 MPa, the material removal depth on Si(100) after ten sliding cycles decreases to the thickness of double atomic layers (Supplementary Fig. 6), which corresponds to the resolution limit of the TEM imaging of the crystalline lattice perpendicular to the Si(100) surface (Fig. 2b).

**Mechanism of ultra-precision fabrication via mechanochemical removal.** During the contact scanning with the $SiO_2$ counter-surface, the atomic layer removal of the Si(100) surface occurs only when water molecules are adsorbed on the surface from the gas phase. In vacuum (less than 10$^{-6}$ Torr), no topographic change was observed after repeated scanning with the $SiO_2$ sphere at a contact stress of 247 MPa. When the contact stress was increased to 631 MPa ($F_n$ was 3 μN), the surface protrusion (which looks like a hillock) was observed in the scanned area, rather than the topographic depression (Supplementary Fig. 7). Similarly, the surface protrusion was observed after scanning in nitrogen, oxygen, or dry air[30]. Thus, the atomic layer removal of the Si(100) surface by contact scan of the $SiO_2$ counter-surface must be due to the mechanochemical reaction involving the adsorbed water molecules at the topmost layer exposed to the gas phase[36].

Not only the adsorbed water, but the chemistry of the counter-surface also plays a significant role in the mechanochemical nanofabrication process[30]. When the counter-surface is replaced with a diamond tip for scanning in humid air, a hillock is formed on the Si surface (Supplementary Fig. 8), instead of a trench. The cross-sectional TEM analysis of the region scratched by the diamond tip reveals that the hillock region is amorphous and the subsurface is plastically deformed (Supplementary Fig. 9). The hillock formation is mainly due to mechanical stress, rather than mechanochemical reactions[35,37]. If the contact stress increases to around 13 GPa, then the surface is plastically deformed producing a groove (topographic depression) (Supplementary Figs. 10 and 11).

In summary, the silicon atomic removal at low scanning loads occurs only when a chemically-reactive counter-surface is used in moisture environments, indicating that both absorbed water

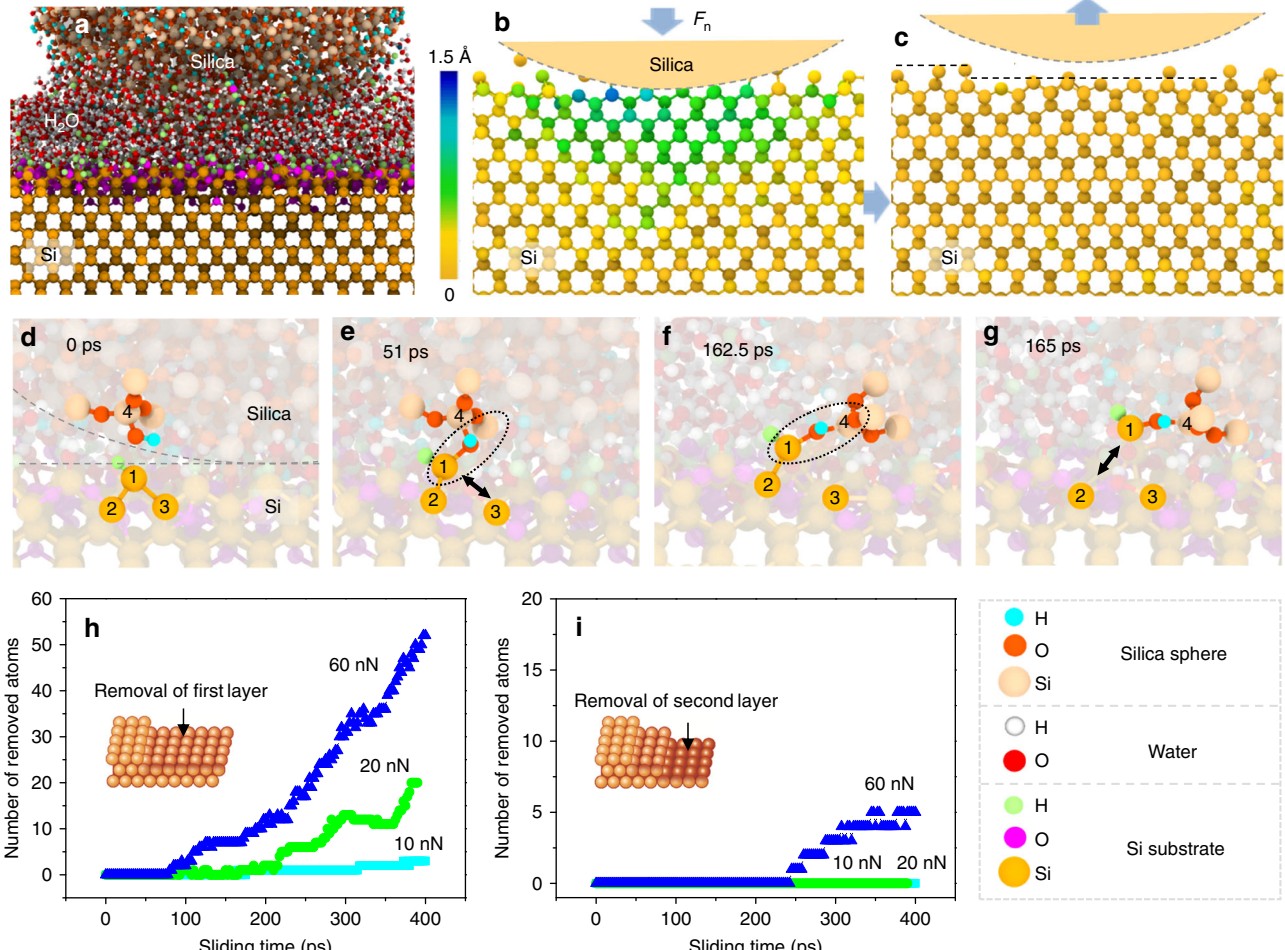

**Fig. 3** Atomic removal of Si surface against silica nano-sphere in MD simulation. **a** Sliding model in MD simulation. **b** Deformation of Si substrate under load condition (The $Z$ direction displacement of Si atoms movement under contact pressure is denoted by different colors). **c** Removal of Si atoms and release of substrate deformation after unloading. **d–g** Mechanochemical reaction process: **d** Initial contact interface reacting with water to form Si–H and Si–OH groups (i.e., $Si_{2-3} = Si_1$–H on down Si substrate and O–$Si_4$–OH on upper $SiO_2$ nano-sphere); **e** Formation of $Si_1$–O–$Si_4$ interfacial bond bridge between Si substrate and silica surfaces, and breaking of $Si_1$–$Si_3$ bond; **f** Tensile stress transferred across $Si_1$–O–$Si_4$ bonding bridges; **g** Fracture of $Si_1$–$Si_2$ bond and the atom $Si_1$ removed from Si substrate. **h** and **i** respectively show the number of Si atoms removed from the first atomic layer (inset schematic in **h**) and second atomic layer (inset schematic in **i**) as a function of sliding time under different load conditions

molecules and chemistry of the counter-surface play critical roles in the tip-based mechanochemical etching of the Si(100) surface. Here a key question is how chemical reactions are activated or facilitated by the interfacial shear. Since the SPM sliding speed is low, the frictional heating of the contact region is negligible; so, thermal reactions can be ruled out. Then, the activation of chemical reaction must originate from the mechanical energy. The mechanical stress dependence can be analyzed by plotting the reaction yield or rate ($\delta$) as a function of applied stress ($\sigma$) in the Arrhenius-type plot[31]

$$\delta = bf_0 \exp\left(-\frac{\Delta U_{act}}{k_B T}\right) \exp\left(\frac{\sigma \Delta V_{act}}{k_B T}\right) \quad (1)$$

Here, $f_0$ is an effective attempt frequency, $b$ a lattice parameter, $\Delta U_{act}$ an activation barrier, $\Delta V_{act}$ the critical activation volume, $k_B$ the Boltzmann's constant, and $T$ the absolute temperature. The dependence of volume loss rate on the applied contact stress follows an exponential relationship (Supplementary Fig. 3b). Since $f_0$ is constant at a given sliding speed, $b$ does not vary unless the substrate is changed, and $\Delta U_{act}$ could be assumed to be constant for a given reaction, the $bf_0\exp(-\Delta U_{act}k_B^{-1}T^{-1})$ term would be constant; then fitting the contact stress dependence data

with Eq. 1 results in an activation volume of 33 $\text{Å}^3$ (Supplementary Fig. 3b). It is noted that this value is comparable to the activation volume (55 $\text{Å}^3$) determined for wear of a Si tip sliding against a polymer surface in humid air[31,33].

Using the threshold contact stress ($\sigma_{min}$) of 247 MPa determined in this experiment (Supplementary Fig. 3b), the critical energy barrier ($\sigma_{min}\Delta V_{act}$) for stress-induced hydrolysis of the Si–Si bonds is estimated as 0.05 eV, which is much lower than the activation energy barrier in vacuum (0.13 eV) estimated from the results in ref. [38]. This indicates that the interfacial water molecule significantly decreases the critical energy barrier for the mechanochemical reaction.

In order to shed light into the mechanism of atomic layer removal of silicon in shear-induced mechanochemical reaction, we carried out molecular dynamics (MD) simulations with a reactive force field called ReaxFF for a $SiO_2$ nano-particle sliding against a silicon substrate in the presence of water molecules (Fig. 3a, for details, see MD simulations section under the Methods section and Supplementary Fig. 12). MD simulations show that the Si substrate elastically deforms under a load of 50 nN exerted by the $SiO_2$ nano-particle (Fig. 3b), and completely recovers, except a few Si atoms removed from the surface, after the load is released (Fig. 3c, Supplementary Movie). The full

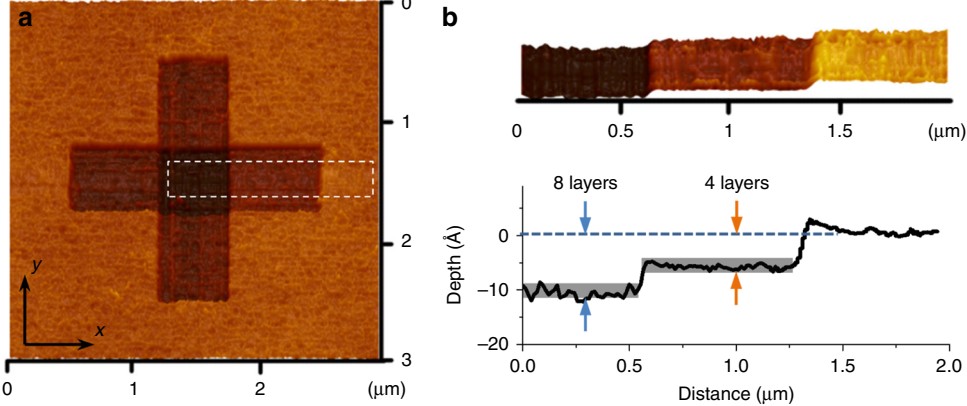

**Fig. 4** Multilayer removals of silicon atoms by scan manufacturing. **a** SPM image of the manufacturing area. **b** 3-D topography and the cross-section profile of the atomic multi-step in **a**. The load is 1 μN and sliding speed is 4 μm s$^{-1}$. In **a**, the first scan was along the horizontal direction (64 lines in 0.5 μm increment along y-axis) and the second scan was along the vertical direction (64 lines in 0.5 μm increment along x-axis)

recovery of the crystalline lattice structure without subsurface damage after the sliding process is consistent with the result observed in HRTEM analysis (Fig. 2, Supplementary Figs. 5 and 6).

MD simulations provide a further insight into the contact behavior and material removal at the atomic level[39,40]. The critical processes for mechanochemical removal of atomic layer could be conceived as three stages (Fig. 3d–g): (i) generation of surface hydroxyl species by reaction of surface atoms with water molecules (Fig. 3d), (ii) formation of interfacial bridge bonds via dehydration reaction between two surface hydroxyl groups across the interface (Fig. 3e), and (iii) dissociation of the substrate bonds under the mechanical shear action (Fig. 3f, g), leading to the removal of a Si atom from the substrate. During the third stage, water molecules impinging from the gas phase further react with the newly exposed surface atoms to form Si–OH groups. Chemical analysis of the mechanochemically-etched area using scanning Auger electron spectroscopy (Supplementary Fig. 13) further supports this mechanism. The absence of mechanochemical reaction in Si/diamond system in humid air (Supplementary Figs. 8–11) indicates that the elastic deformation alone cannot induce dissociation of the Si–Si bonds; it seems that the formation of interfacial bridge bonds between sliding solid surfaces is necessary. Therefore, only the Si atoms in the first layer interacting with the adsorbed water molecules and the counter-surface can readily undergo chemical reactions and be removed preferentially. As shown in Figs. 3h and i, the Si atom removal is limited to the first layer at a load of 20 nN. A lower load (e.g., 10 nN) is insufficient to induce Si atomic removal, while a higher load (e.g., 60 nN) can cause the loss of Si atoms in the second layer.

**Controlled nanomanufacturing with atomic layer removal.** By altering the mechanochemical reaction conditions (i.e., via adjusting mechanical stress or scanning cycles), one can achieve controlled removal of the multiple atomic layers of Si. Fig. 4 shows the SPM image of the area scanned twice—first in the horizontal direction and second in the vertical direction—at a contact stress of 428 MPa ($F_n$ was 1 μN). The depth of atomic layer removed per single cycle of sliding amounts to around 5.5 Å (corresponding to four atomic layers). The center region scanned twice in orthogonal directions shows a removal depth corresponding to eight atomic layers (Fig. 4b).

**Discussion**
In summary, a SPM tip-based, mask-free and chemical-free lithographic process producing topographic features into the Si

(100) surface was demonstrated. This process is based on shear-induced mechanochemical reactions involving silicon atoms at the topmost surface of the substrate, water molecules adsorbed from the ambient air, and hydroxyl groups at the counter-surface. By controlling the contact scan condition, it is possible to attain the precision down to removal of single atomic layer of silicon which would be the ultimate resolution in the depth direction in topographic patterning on the Si(100) surface. Because the mechanochemical reaction is limited at the topmost surface only at mild shear conditions, it allows the removal of single atomic layer from the scanned area without any subsurface damages (plastic deformation or lattice defects). This mechanochemistry-associated manufacturing approach might be applicable to other substrates, such as, GaAs for fabrication of the site-controlled nanopatterning[41,42] and 2D materials for layered removal[43]. This study opens up a new opportunity for achieving the ultimate precision nanofabrication and reveals the potential for combining the mechanochemistry and SPM scanning to advance the ultra-precision nanomanufacturing processes.

**Methods**
**Material preparation.** The native oxide of a Si(100) wafer was removed by immersing the wafer in a 40% aqueous solution of hydrofluoric acid for 3 min followed by rinsing in methanol, ethanol, and deionized water with sonicating in each solvent. This produced a hydrophobic surface passivated with hydrogen atoms.

**Manufacturing tests with SPM.** The surface manufacturing and in situ topography scanning were performed with SPM. In the scanning-fabrication mode, the Si wafer was rubbed with a SiO₂ microsphere (diameter was 2.5 μm) attached to an SPM cantilever (Supplementary Fig. 1). The normal spring constant of the cantilever was calibrated as 12.1 N m$^{-1}$. If not specially mentioned, the scanning area was 1 μm × 1 μm (256 lines μm$^{-1}$); scanning speed $v$ was 4 μm s$^{-1}$. The load $F_n$ ranged from 300 to 3100 nN ($F_n$ are the sum of $F_{applied}$ and $F_{adhesion}$, where adhesion force $F_{adhesion}$ was about 50 nN in humid air with the relative humid (RH) of 75 ± 2%) and the contact stress was estimated to be from 247 to 593 MPa by the following equation[38].

$$\sigma = \frac{F_n}{A_{contact}} \qquad (2)$$

Here, $A_{contact}$ is the area of contact calculated using the DMT model. After manufacturing, surface topography was imaged in vacuum (less than 10$^{-3}$ torr) with a sharp Si₃N₄ tip with a radius less than 20 nm and a nominal spring constant of 0.1 N m$^{-1}$. The scanning precision of SPM in vertical direction was calibrated by scanning a step of single layer graphite. As shown in Supplementary Fig. 2, the height of single layer graphite was measured as 3.4 ± 0.2 Å. The theoretical height of single layer is 3.4 Å, confirming the accuracy of our system in depth measurement.

**MD simulations**. Prior to the sliding process in MD simulations, the Si(100) surface was prepared as follows. First, the Si(100) surface with dimensions of $69.12 \times 69.12 \times 30.09$ Å (24 Si atomic layers with 7776 atoms) was equilibrated to 300 K using the Nose-Hoover thermostat for 20 ps with a temperature damping constant of 0.025 ps. Then it was relaxed for 10 ps. Finally, the Si(100) surface was used to react with 8000 $H_2O$ molecules for 100 ps under 300 K to obtain the chemical state of Si surface close to that in real case, where the surface was terminated by Si–H, Si–OH, and Si–O–Si functional groups. The Si(100) substrate was divided into three layers, including bottom-most fixed layer, thermostat layer in the middle, and free layer at the top (Supplementary Fig. 12). The hemispherical $SiO_2$ tip (radius was 30 Å), with inverse order of the three layers (Supplementary Fig. 12), was cleaved from the initial amorphous silica structure produced from a melt quench process of a bulk quartz silica crystal, and it was fully terminated with hydroxyls. A liquid layer with 1600 water molecules (around 1.0 nm thickness) was constructed to cover the Si substrate (Supplementary Fig. 12). The simulations were performed using the LAMMPS code[44]. A ReaxFF force field was used to describe the interaction between Si, $SiO_2$, and water; the more details of this force field can be found in refs.[45,46]. Periodic boundary conditions were applied to both $x$ and $y$ directions to mimic laterally infinite surface. A time step of 0.25 fs was used with the Velocity Verlet algorithm to integrate the equations of motion. Normal loads of 10, 20, 50, and 60 nN were applied uniformly on the rigid layer of the silica sphere, then the tip was slid with a constant velocity of 10 m s$^{-1}$ along the sliding direction ($x$ direction). In order to control the system temperature, the thermostat layers were coupled to a Langevin thermostat method with a temperature damping constant of 100 fs. The remaining atoms of the free layers and water layer were free of constraints so that they moved according to the interatomic forces.

**Data availability**. The data which supports the findings of this work is available upon request from the corresponding author.

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

## Acknowledgements

We acknowledge the support from the Natural Science Foundation of China (91323302, 51527901, 51505391 and 51375010). S.H.K. acknowledges the support from the National Science Foundation of the USA (grant No. CMMI-1435766). L.M.Q. acknowledges the Science Challenge Program of China (No. TZ2018006). L.C. acknowledges the Fundamental Research Funds for the Central Universities (2682016CX026). Simulations were carried out on the 'Explorer 100' cluster system of the Tsinghua National Laboratory for Information Science and Technology.

## Author contributions

L.C., L.M.Q., and S.H.K. conceived and designed the experiments; L.C., P.Z., B.J.Y. and C.C. performed experiments; L.C., J.L.W., T.B.M., S.H.K., and X.C.L. designed the MD

simulation; J.L.W., T.B.M., and X.C.L. carried out the MD simulation; L.C., L.M.Q., J.L. W., T.B.M., and S.H.K. analyzed the data; and L.C., L.M.Q., J.L.W., and S.H.K. wrote the manuscript.

## Additional information

**Competing interests:** The authors declare no competing interests.

