## [Peer Review File · Nature Communications]

Reviewers' Comments:

Reviewer #1:

Remarks to the Author:

The authors have provided an experimental and simulation work on material removal at the nano-level, for silicon.

The paper is interesting but several revisions need to be made before publication.

The authors use the term "ultimate-precision" in the title, while ultra-precision is used throughout the paper. The second expression is more frequently used. However, "nanomanufacturing" implies ultra-precision, in my opinion. I believe that the authors should revise the title of their paper. Furthermore, "Molecular Dynamics" is more appropriate than "Molecules Dynamics" used by the authors.

The abstract of the paper should be rewritten to summarize the main findings of the analysis presented in the paper.

In the introductory section, the authors must exhibit the novelty of their work by presenting the features of previous works and commenting on the improvements they made. Instead, most of the references are old and not adequately discussed.

The authors state that 247 MPa is the critical contact stress, in page 5. Please, justify this statement more.

Furthermore, it is stated that "additional experimental evidences...plastic flow or fracture". Please, elaborate more.

In page 8 it is written that "the results...mechanochemical reaction". This is not adequately specified. Please elaborate more.

Finally, more details on MD simulation, e.g. potential functions used, time-step etc. should be provided.

Reviewer #2:

Remarks to the Author:

The paper describes a novel mechanochemical mediated reaction between a colloidal SPM tip and a silicon surface in the presence water molecules absorbed from the ambient environment. The authors present compelling experimental observations that are backed up with detailed theory, primarily in the form of molecular-dynamic simulations. These observations lead the authors to conclude that their approach enables the mechanochemical removal of materials from a Si surface, with sub-nm depth control and negligible damage to the substrate. The data and analysis presented are convincing within the constraints of the available information.

This is of interest to researchers in the field as it is clearly distinct from other SPM lithography approaches employed on silicon that either employ intermediate 'resist' layers or rely on 'additive' processes such as tip-induced oxidation or dip-pen lithography. If reproduced by other groups, it will offer a new variation on existing SPM lithography techniques.

As stated above, the claims are convincing within the constraints of the available information. The data provided is appropriate and strong enough for such preliminary research. However, an even more compelling case could have been made with the inclusion of surface analysis, such as Scanning Auger or XPS, which would further support the proposed mechanochemical mechanism. These techniques are not intrinsically difficult, but do require access to specialized, expensive equipment.

The main weakness in the manuscript stems from its presentation of the technique as achieving "Ultimate-precision", the "ultimate limit of manufacturing" and demonstrating "ultimate resolution", whilst stating that "ultimate nanomanufacturing precision for crystalline silicon without subsurface structural damage has not been reported". These statements are hyperbole at best and misleading at worst.

In the field of nanofabrication, damage-free lithographic patterns in silicon down-to and even below 10 nm lateral resolution are considered relatively routine. When combined with atomic layer etching (ALE), these approaches match and exceed the resolution and precision reported in this work. Indeed, modern electronic and opto-electronic devices would not function correctly if this were not the case.

In addition to practically neglecting the mature field of nanofabrication, the authors also skim over the field of Scanned Probe Lithography as “Recently, atomic force microscopy (AFM) has been demonstrated as an alternative means for high-precision material removal...”. There is literature on this technique dating back to around the year 2000 (see e.g. descriptions of AFM tip induced electrochemical oxidation of Si {Journal of Vacuum Science & Technology A: Vacuum, Surfaces, and Films 15, 1451 (1997)} or mechanochemical oxidation of Si {Jpn. J. Appl. Phys. Vol. 40 (2001) pp. L 1247–L 1249}. Both the approaches presented in the examples above have enough similarity to the work presented in this manuscript to warrant a serious mention in the introduction. It should also be noted that each of them also report better lateral resolution than the work in this manuscript.

In conclusion, I believe the work reported in this paper is clearly presented and novel. However, I also feel that the title, abstract, introduction and conclusions need significant modification to 1) place the work more clearly in the context of current nanofabrication approaches, 2) provide a more accurate representation of previous scanned probe lithography work, which dates back well over a decade and 3) moderate the claims of the technique being “ultimate” in the light of points 1 and 2 above.

PennState

Department of Chemical Engineering
Materials Research Institute
N-323 Millennium Science Complex
University Park, PA 16802-2150

Seong H. Kim, Dean's Fellow, Professor
Ph one: (814) 863-4809
Fax: (814) 865-7846
E-mail: shkim@enr.psu.edu

Re: Revision of manuscript NGOM-15-1730236

Dear Dr. Stuart Thomas,

It was very helpful to get thorough and critical review comments. We believe that this is an indication that the reviewers felt that the experimental data and mechanisms presented in this manuscript are of great interest to the readers of *Nature Communications*. We have addressed all comments raised by two reviewers. Two additional files, including revised manuscript with marked changes and manuscript change list, were submitted as Related Manuscript files to help checking the modifications. Below are the summary of the point by point responses and revisions. With this revision, the manuscript is considerably improved and we believe that it is suitable for publication. We look forward to hearing of your positive comment and decision.

Best regards,

Seong H. Kim

Formatted: Font: 11 pt, Font color: Black

Formatted: Font: 11 pt, Font color: Black

Formatted: Font: 11 pt

Response to Reviewer #1:

The authors have provided an experimental and simulation work on material removal at the nano-level, for silicon.

The paper is interesting but several revisions need to be made before publication.

- *The authors use the term "ultimate-precision" in the title, while ultra-precision is used throughout the paper. The second expression is more frequently used. However, "nanomanufacturing" implies ultra-precision, in my opinion. I believe that the authors should revise the title of their paper. Furthermore, "Molecular Dynamics" is more appropriate than "Molecules Dynamics" used by the authors.*

→ Thanks for the positive comment and suggestion. We used the term "ultimate-precision" to emphasize that the described process could achieve the removal of surface atoms with a control down to the single atomic layer. We admit that the use of "ultimate-precision" without properly describing or clearly defining it could cause confusions or misconceptions. We have addressed this issue by providing its definition in the main text. The title was revised to "Nanomanufacturing of silicon surface with a single atomic layer precision via mechanochemical reactions". Also, we reduced redundant usages of the ultimate-precision or ultra-precision terms in the manuscript.

→ The error of calling Molecules Dynamics, instead of Molecular Dynamics, was corrected. We also read the manuscript carefully and made additional typographical changes / corrections.

- *The abstract of the paper should be rewritten to summarize the main findings of the analysis presented in the paper.*

→ Following the reviewer's suggestion, we rewrote the abstract focusing on the scientific findings and their implications in nanomanufacturing applications.

- *In the introductory section, the authors must exhibit the novelty of their work by presenting the features of previous works and commenting on the improvements they made. Instead, most of the references are old and not adequately discussed.*

→ The second review made a similar comment; we have substantially changed the content in the introduction section, commenting various previous literatures and citing more updated references.

- *The authors state that 247 MPa is the critical contact stress, in page 5. Please, justify this statement more.*

→ To explain this issue, we modified Figure S3 in the Supplementary Materials. Basically, the contact stress of 247 MPa was the lowest value that created any measureable surface-etch while the contact load was increased step-wise in experiment. Thus, it could be taken as the upper-limit of the critical contact stress. The actual value might be slightly lower than 247 MPa. This was elaborated in the manuscript.

- *Furthermore, it is stated that "additional experimental evidences...plastic flow or fracture". Please, elaborate more.*

→ SPM image shown in Figure 1 demonstrates the removal of a single atomic layer from the Si surface – the etch depth is very close to the theoretical thickness of Si monolayer, and the surface roughness of the etched region is similar to that of the pristine region. A few hypotheses could be posited – whether the depression in the AFM scanned region is due to purely mechanical processes (which would be accompanied with plastic flow or fracture) or mechanochemical reactions of the top-most layer involving reactions of water molecules adsorbed from the gas phase. To answer this question, cross-sectional imaging with HRTEM was carried out. This was clarified in lines 92-95 in the revised manuscript.

- *In page 8 it is written that "the results..mechanochemical reaction". This is not adequately specified. Please elaborate more.*

→ In the section immediately following the sentence presenting possible hypotheses (line 75 of the original manuscript), we have discussed that the purely mechanical process can be ruled out. Then, several control experimental data were presented, which clearly indicated that the obtained nanomanufacturing results must originate from chemical reactions involving the adsorbed water molecules and the atoms or functional groups of the counter-surface under interfacial shear conditions. To describe the mechanochemical mechanism more clearly, this sentence has been modified as “*In summary, silicon atomic removal at extreme low scanning load only occurs against an active counter-surface in moisture environment, indicating that both adsorbed water molecules and chemistry of the counter-surface play critical roles in the tip-based mechanochemical etching of the Si(100) surface.*” (lines 147-149 in the revised manuscript).

- *Finally, more details on MD simulation, e.g. potential functions used, time-step etc. should be provided.*

→ The potential function we used in the simulations is a ReaxFF reactive force field, and the time-step is

0.25 fs for all the simulations (which is typical in ReaxFF-MD). These were added in the revised manuscript (Please see **MD simulation method in the main text**).

Response to Reviewer #2:

- *The paper describes a novel mechanochemical mediated reaction between a colloidal SPM tip and a silicon surface in the presence water molecules absorbed from the ambient environment. The authors present compelling experimental observations that are backed up with detailed theory, primarily in the form of molecular-dynamic simulations. These observations lead the authors to conclude that their approach enables the mechanochemical removal of materials from a Si surface, with sub-nm depth control and negligible damage to the substrate. The data and analysis presented are convincing within the constraints of the available information.*

→ We appreciated the positive comment about the experimental and computational results presented in this manuscript.

- *This is of interest to researchers in the field as it is clearly distinct from other SPM lithography approaches employed on silicon that either employ intermediate ‘resist’ layers or rely on ‘additive’ processes such as tip-induced oxidation or dip-pen lithography. If reproduced by other groups, it will offer a new variation on existing SPM lithography techniques.*

→ The process is reproducible as long as the silicon substrate and AFM tip surfaces are free of contaminants as well as the relative humidity and applied load are controlled. We appreciated this reviewer’s point contrasting our approach with other SPM-based lithography. We have incorporated the comparison with other SPM lithography techniques into the introduction section of the revised manuscript.

- *As stated above, the claims are convincing within the constraints of the available information. The data provided is appropriate and strong enough for such preliminary research. However, an even more compelling case could have been made with the inclusion of surface analysis, such as Scanning Auger or XPS, which would further support the proposed mechanochemical mechanism. These techniques are not intrinsically difficult, but do require access to specialized, expensive equipment.*

→ The techniques that the reviewer mentioned are not intrinsically difficult; however, applying them to characterization of the mechanochemically-reacted area is not easy because the area is so small. In our case, it was only a few microns wide and less than one nanometer deep with very little chemical contrast. Indeed, the suggested experiment was extremely difficult; but, after many trials, we were finally able to get the data that supports the mechanochemistry discussed in the main text.

→ We have chosen scanning Auger electron spectroscopy (s-AES) over x-ray photoelectron spectroscopy (XPS) since s-AES has a better spatial resolution and a better surface sensitivity than XPS. First, we created two scratch lines using a diamond tip that were deep enough to be found easily with secondary electron microscopy unit equipped in our s-AES system. Then, between the two deep scratch lines, we produced a $7\ \mu\text{m} \times 7\ \mu\text{m}$ area with a $\sim 2\ \text{nm}$ depth through the same mechanochemical etch process described in the manuscript (**Fig. S13a** and **S13b** in the Support Information). Using the scratch lines as a reference point, we were able to find the mechanochemically-etched region. Then, s-AES analysis was performed AES analysis for the mechanochemically-etched region and the surrounding area (**Fig. S13c**). We also analyzed the “freshly HF-etched” Si(100) wafer surface (which was the substrate studied) and the “as-received” Si(100) wafer surface (covered with native oxide layers) for comparison. **Fig. S13d** compares the relative intensity of oxygen signal with respect to the silicon signal.

Note that although the HF-etch completely removes the native oxide on the silicon wafer, the AES spectrum of the freshly HF-etched surface still contains a small amount of oxygen ($I_{\text{O}}/I_{\text{Si}} \approx 0.17$) because

of the natural oxidation process occurring during the sample transfer and handling in ambient air. The sample tested for the mechanochemical nano-patterning with AFM was exposed to air for a longer time than the freshly HF-etched surface; thus, it has a larger oxygen content ($I_{O}/I_{Si} \approx 0.22$; this is inevitable due to the thermodynamics of surface oxidation of a clear silicon wafer). ***The important finding here is that the mechanochemically-etched region has the oxygen content slightly larger ($I_{O}/I_{Si} \approx 0.38$) than the intact region outside the patterned region.*** This is the consequence of mechanochemical reactions involving adsorbed water molecules at the topmost surface upon shear with the counter-surface.

- *The main weakness in the manuscript stems from its presentation of the technique as achieving “Ultimate-precision”, the “ultimate limit of manufacturing” and demonstrating “ultimate resolution”, whilst stating that “ultimate nanomanufacturing precision for crystalline silicon without subsurface structural damage has not been reported”. These statements are hyperbole at best and misleading at worst.*
 - We admitted that emphasizing the “ultimate-precision” too much without properly defining it could give an impression of exaggeration or over-claim. In this paper, we meant to highlight the attainment of the single atomic layer removal in a designated area of a silicon wafer without using any chemicals or sacrificial masks. We have toned down the use of the “ultimate” term throughout the manuscript and added more scientific clarity into the smallest layer thickness that can be removed by any etching processes.
- *In the field of nanofabrication, damage-free lithographic patterns in silicon down-to and even below 10 nm lateral resolution are considered relatively routine. When combined with atomic layer etching (ALE), these approaches match and exceed the resolution and precision reported in this work. Indeed, modern electronic and opto-electronic devices would not function correctly if this were not the case.*
 - We agree that in hindsight, we overlooked other chemical processes such as ALE (which uses sequential, self-limiting thermal reactions and achieve the reverse of atomic layer deposition, ALD). The main advancement or difference of the process described in this manuscript is that, unlike ALE, it does not require any sacrificial layers for nano-patterning, does not require any harmful chemicals, and can be done in ambient conditions. This is properly highlighted in the revised introduction section.
- *In addition to practically neglecting the mature field of nanofabrication, the authors also skim over the field of Scanned Probe Lithography as “Recently, atomic force microscopy (AFM) has been demonstrated as an alternative means for high-precision material removal...”. There is literature on this technique dating back to around the year 2000 (see e.g. descriptions of AFM tip induced electrochemical oxidation of Si {Journal of Vacuum Science & Technology A: Vacuum, Surfaces, and Films 15, 1451 (1997)} or mechanochemical oxidation of Si {Jpn. J. Appl. Phys. Vol. 40 (2001) pp. L 1247–L 1249}. Both the approaches presented in the examples above have enough similarity to the work presented in this manuscript to warrant a serious mention in the introduction. It should also be noted that each of them also report better lateral resolution than the work in this manuscript.*
 - Thanks for suggesting these references. They have been added, along with other recent papers, in the revised manuscript.
- *In conclusion, I believe the work reported in this paper is clearly presented and novel. However, I also feel that the title, abstract, introduction and conclusions need significant modification to 1) place the work more clearly in the context of current nanofabrication approaches, 2) provide a more accurate representation of previous scanned probe lithography work, which dates back well over a decade and 3) moderate the claims of the technique being “ultimate” in the light of points 1 and 2 above.*

→ All sections suggested (title, abstract, introduction, and conclusions) have been substantially modified for clarity and accuracy of the literature citing. In addition, the “ultimate precision” term is defined clearly and its usage has been reduced.

Reviewers' Comments:

Reviewer #1:

Remarks to the Author:

The authors have adequately addressed the reviewers' comments and have improved their manuscript. The manuscript can be published as is.

Reviewer #2:

Remarks to the Author:

It's pleasing to see that the authors have taken all of the comments seriously, addressing each in turn and clearly articulating their changes in the associated documents. In particular, the effort expended on undertaking additional experiments, specifically sAES, is to be commended. Having reviewed these changes, I am happy with their modifications and agree that they have revised their manuscript to the scientific standard required for publication. However, I feel that the document, particularly the newly added sections, requires a thorough proof-read. It contains a significant number of minor but distracting grammatical errors such as the incorrect placement of 'a' and 'the' and I feel that eliminating these would further enhance the document.

I therefore recommend publication based on the quality of the scientific content. However, I also make the strong suggestion that the document is thoroughly proof-read prior to publication.

Response to the review comments:

Reviewer #1: *The authors have adequately addressed the reviewers' comments and have improved their manuscript. The manuscript can be published as is.*

Reviewer #2: *It's pleasing to see that the authors have taken all of the comments seriously, addressing each in turn and clearly articulating their changes in the associated documents. In particular, the effort expended on undertaking additional experiments, specifically sAES, is to be commended. Having reviewed these changes, I am happy with their modifications and agree that they have revised their manuscript to the scientific standard required for publication. However, I feel that the document, particularly the newly added sections, requires a thorough proof-read. It contains a significant number of minor but distracting grammatical errors such as the incorrect placement of 'a' and 'the' and I feel that eliminating these would further enhance the document. I therefore recommend publication based on the quality of the scientific content. However, I also make the strong suggestion that the document is thoroughly proof-read prior to publication.*

→ Thanks for your suggestions. The manuscript was thoroughly proof-read and minor grammatical errors were all corrected.